# Oral lactoferrin administration does not impact the diversity or composition of the infant gut microbiota in a Peruvian cohort

Luis González,[1,2] Jose Luis Paredes Sosa,[1] Susan Mosquito,[1] Yesenia Filio,[1] Pedro E. Romero,[3] Theresa J. Ochoa,[2] Pablo Tsukayama[1,2,4]

**ABSTRACT**  Lactoferrin is a protein from breast milk involved in early-life microbiome assembly. It is believed to promote the growth of lactic acid-producing bacteria and inhibit pathogens by limiting iron availability and enabling an acidic environment in the gut. Previous reports suggest that oral administration of lactoferrin in neonates promotes the increase of gut microbiota diversity and the enrichment of beneficial microbes. However, its effect on infant gut microbiota over time has yet to be thoroughly studied to fully understand its potential as a prebiotic. Using 16s rRNA gene amplicon sequencing, we analyzed the gut microbiota composition of 60 toddlers at 12–18 months of age from Lima, Peru, who received daily oral administration of lactoferrin or placebo for 6 months since enrollment. Samples were analyzed at 0, 2, and 4 months after beginning treatment. Our results show that lactoferrin treatment does not increase gut microbiota diversity over time nor affect its bacterial composition compared to the placebo group.

**IMPORTANCE**  Previous studies have suggested that oral lactoferrin enhances diversity in the gut microbiota in infants while inhibiting the growth of opportunistic pathogens. However, the effect of lactoferrin on infant gut microbiota over time has yet to be thoroughly studied. Our study suggests that lactoferrin oral treatment in infants aged 12–18 months does not affect gut microbiome diversity and composition over time. To our knowledge, this is the first study to report the effect of lactoferrin on infant gut microbiome composition over time and helps elucidate its impact on infant health and its therapeutic potential.

**KEYWORDS**  lactoferrin, early life microbiome, 16S rRNA sequencing, Peru, gut microbiota

The microbiota is an ecosystem of microorganisms (e.g., bacteria, archaea, viruses, fungi, and microeukaryotes) that contribute to the host's metabolic, immune, and cognitive functions (1, 2). Its composition is influenced by host genetics, diet, environmental exposure, and medical treatments, among other factors (3). Complex and stable microbial communities are found across multiple body sites, although most microorganisms reside in the gut (4, 5). The gut microbiota possesses an enormous functional diversity encoded by a collection of genes (the "microbiome") 10–100 times larger than the human genome. It encodes the synthesis of essential metabolites and functions that benefit its host (4). Hence, alterations in the gut microbiota composition affect its functionality, thereby influencing the host's health status (6).

Neonates partially acquire the maternal microbiota through the birth canal during delivery and through the skin and oral routes during close contact (7, 8). Vaginally delivered breastfeeding infants have been associated with a microbiota dominated by the genus *Bifidobacterium*, which promotes a more acidic environment in the gut, thereby inhibiting the growth of pathogenic bacteria (6). Through early development,

Address correspondence to Pablo Tsukayama, pablo.tsukayama@upch.pe.

The authors declare no conflict of interest.

See the funding table on p. 10.

the gut microbiota provides essential functions such as immune system maturation and resistance to pathogen colonization (4, 6, 9).

Early perturbation of the gut microbiota may affect the neonate's development, influencing their health status and susceptibility to various conditions in later life (e.g., asthma, Crohn's disease, diabetes, milk allergy, etc.) (8, 10). One such disruption is the reduced transmission of the microbiota from the mother to the baby, which could occur through cesarean delivery, formula feeding, and antibiotic exposure (6). For instance, cesarean-delivered babies display a reduction of *Bifidobacterium* and *Bacteroides* genera compared to vaginally delivered neonates (11). Moreover, cesarean-delivered babies show enrichment of opportunistic bacteria (e.g., *Enterococcus*, *Enterobacter*, and *Klebsiella*) compared to vaginally delivered neonates (11). In contrast, formula-fed infants present an enrichment of opportunistic bacteria (e.g., *Clostridioides difficile*, *Streptococcus faecalis*, *Pseudomonas aeruginosa*, and *Enterococcus faecalis*) compared to breastfed infants (7). Although perturbation in cesarean-delivered infants can be restored through breast milk consumption, it results in lower diversity than breastfeeding, vaginally delivered neonates (12). Gut microbiota perturbation favors the growth of opportunistic pathogens in the gut (4), which represents a significant risk since infants do not possess a mature immune system (6).

Thanks to an improved understanding of early-life microbiota assembly dynamics and associations with childhood diseases, new preventive treatments have been developed to address microbial imbalances in the early-life gut microbiota. For instance, probiotics (e.g., *Bifidobacterium*, *Lactobacillus*) and prebiotics have been used to restore the perturbed gut microbiota (13). Moreover, abnormal microbiota of infants delivered by cesarean section can be partially restored by orally delivered fecal microbiota transplants from the mother (14). However, this represents a risk since the mother can transfer opportunistic pathogens to the neonate, whose immature immune system may be less capable of handling opportunistic colonization (6).

Breastfeeding is essential for the normal development of infant microbiota (15). Lactoferrin is an iron-binding protein in breast milk that reduces iron availability, inhibiting the growth of enteric pathogens (e.g., *Listeria monocytogenes*, *Salmonella enterica*, and *Escherichia coli*) and their capacity to form biofilms (16, 17). Lactoferrin represents approximately 20% of the total proteins in breast milk (16). Interestingly, it does not appear to inhibit the growth of probiotic bacteria *in vitro* and *in vivo* (18, 19). A higher concentration of fecal lactoferrin in neonates has been associated with a greater abundance of bifidobacteria and lactobacilli in infants' feces (19, 20). A clinical trial using talactoferrin, a recombinant human lactoferrin, has shown positive effects on the gut microbiota of low-birth-weight neonates when supplemented orally twice daily (21). Furthermore, lactoferrin has shown a beneficial impact on neonates and infants by reducing rates of sepsis, necrotizing enterocolitis, and diarrhea (21–24).

Lactoferrin's antimicrobial and bacteriostatic properties have been assessed *in vitro* (21), and previous reports suggest that oral lactoferrin administration enhances diversity in the gut microbiota in neonates (7, 21). A richer microbial diversity is correlated to good health outcomes since it allows the microbiota to have a richer functionality, which the host can take advantage of Ghosh and Pramanik (25). Nevertheless, the effect of lactoferrin on infant gut microbiota over time has yet to be fully described; hence, we cannot fully understand its role in microbiota development and its therapeutic potential during infancy. By analyzing samples from a previous lactoferrin clinical trial, this study describes the effect of daily lactoferrin supplementation on the gut microbiota diversity and composition of toddlers 12–18 months of age from a peri-urban community in Lima, Peru (24).

## RESULTS

### Study background

Samples used in this study belong to a randomized, double-blind controlled trial of bovine lactoferrin to prevent diarrhea in children (24). Specimens were collected from July 2008 to May 2009 from toddlers 12–18 months old (Table S1). Eligible participants were previously weaned at 12–18 months old to evaluate the oral daily dose of lactoferrin's effect in preventing diarrhea exclusively. Exclusion criteria were a history of severe, persistent, or chronic diarrhea, severe malnutrition, serious infections requiring hospitalization in the month prior, severe chronic illness, or a personal or family history of allergy to cow's milk or infant formula, eczema, allergic rhinitis, or asthma (24). Five hundred fifty-five infants participated in the original study; 278 received lactoferrin, and 277 received a placebo for 6 months (Fig. 1A). Participants received 0.5 g twice daily lactoferrin or placebo (maltodextrin) diluted in 25 mL of water.

We selected 60 participants, of which 31 received lactoferrin (15.59 +/− 1.9 months old), and 29 received placebo (15.93 +/− 2.2 months old). We analyzed three stool samples from each patient ($n = 180$) at three time points: 0 (M0), 2 (M2), and 4 (M4) months after the start of the treatment (Fig. 1B). At the collection point, M0, patients had not started the treatment. To ensure both experimental groups (LF and PB) started the treatment with the same clinical conditions, we compared the following variables: weight, BMI, ZWAS, and ZBMI (Table S1). They were no statistical significance in the clinical covariables among experimental groups, except when comparing the BMI (Table S2). To assess if these covariables might interfere with the assessment of lactoferrin over the gut microbiota, we performed a permanova analysis to measure the effects of the clinical covariables in microbiota composition; however, we did not observe a statistically significant impact of these clinical covariables in the gut microbiota (Table S3).

### Effect of lactoferrin and placebo treatment on gut microbiome diversity and composition

We performed 16s rRNA amplicon sequencing on 180 samples, of which 172 passed QC (Table S2). We observed many sequences identified as chimeras; however, samples that passed QC were good enough for describing gut microbiota composition. Up to 197 unique bacterial genera were found in our data set. The most abundant genera identified, independently of treatment and sample collection point, were *Bifidobacterium*, *Blautia*, *Streptococcus*, *Erysipelotrichaceae UCG-003*, and *Akkermansia* (Fig. S1).

To assess gut microbiota diversity among experimental groups (lactoferrin vs placebo) over time (M0, M2, and M4), we used observed amplicon sequence variants (ASVs), Shannon, and Inverse Simpsons' alpha diversity metrics. We did not find a significant difference in gut microbiota diversity among experimental groups at any time (Fig. 2; Table S3).

We performed beta diversity analysis using the Aitchison distance metric (26) to compare gut microbiota composition similarity among experimental groups depicted in a dendrogram tree (Fig. 3A). We found instances in which all samples from the same individual clustered in the dendrogram (e.g., samples from patients 14 and 26 [P14 and P26]), showing that they share a similar gut microbiota composition. However, samples are more often clustered with those from different individuals. On the other hand, samples did not cluster according to the patient's treatment (lactoferrin/placebo) (Fig. 3A).

We plotted our beta diversity results in a principal coordinate analysis (PCoA) graph to assess the similarity of gut microbiome composition among experimental groups (Fig. 3B). We compared microbiota composition at the start (M0) and end (M4) of the treatment among experimental groups (Fig. 3A and B) and also compared microbiota composition among experimental groups at the end of the treatment (Fig. 3B). Samples did not cluster according to their time of collection (Fig. 3A and B) or treatment (Fig. 3B).

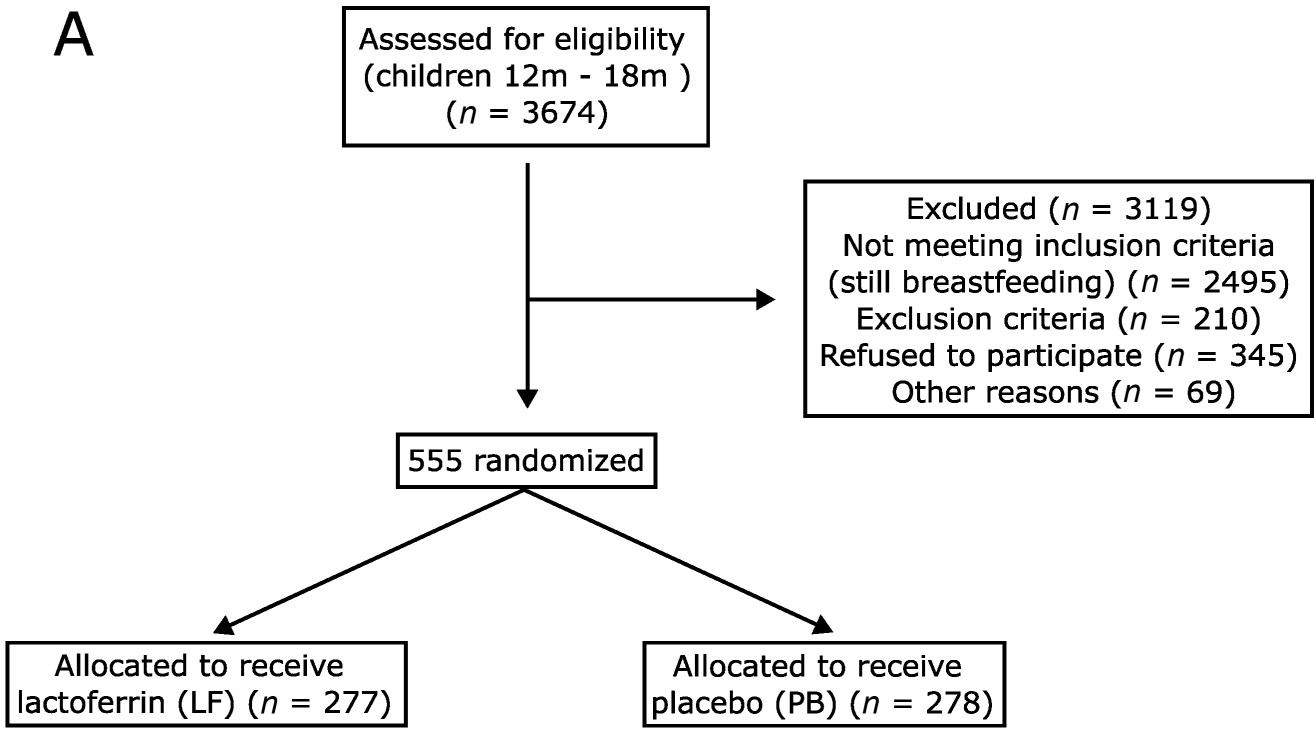

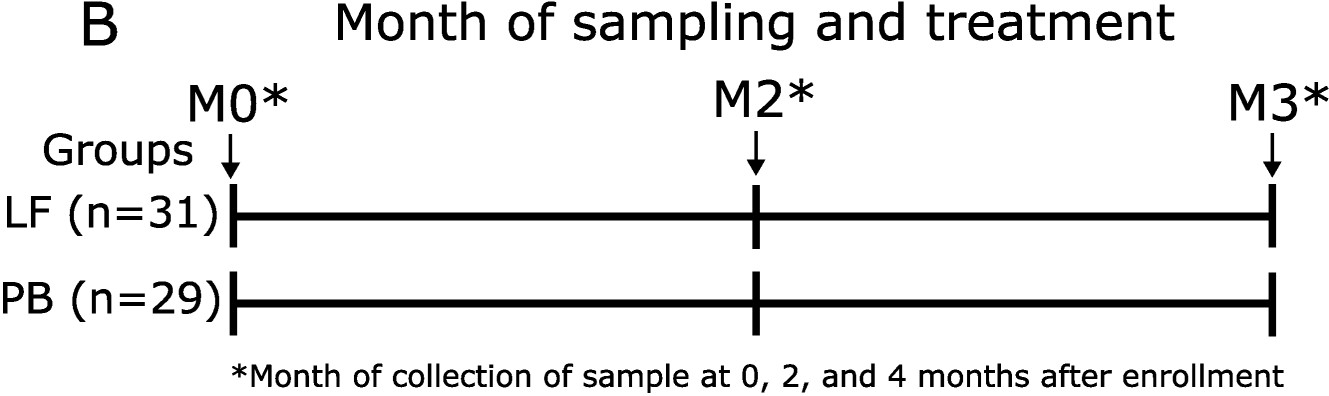

*Month of collection of sample at 0, 2, and 4 months after enrollment

FIG 1 (A) Flow diagram of the original lactoferrin clinical trial. We selected samples from 60 participants, of which 31 received lactoferrin (LF) and 29 received placebo (PB). (B) Diagram of sample collection time point. We sequenced three stool samples per patient at 0 (**M0**), 2 (**M2**), and 4 (**M4**) months after the treatment.

Detailed statistical information is available in the Supplementary Materials section (Table S4).

### Effect of lactoferrin oral treatment in specific taxa enrichment

We used LEfSe (27) to identify enriched genera in the microbiota among each experimental group at different time points (Fig. 4; Fig. S2; Table S5). Since this study only includes 16S analysis, we can only identify bacterial taxa at the genus level. We compared the microbiome composition of both groups at M0 to ensure both experimental groups started with the same conditions at the start of the trial. We found an enrichment of *Prevotella*, *Megasphaera*, and *Roseburia* in group PB-M0 compared to LF-M0 (Fig. S3). However, when calculating the Aitchison distance metric, we didn't find significant global differences in the microbiome composition between groups (Fig. S4).

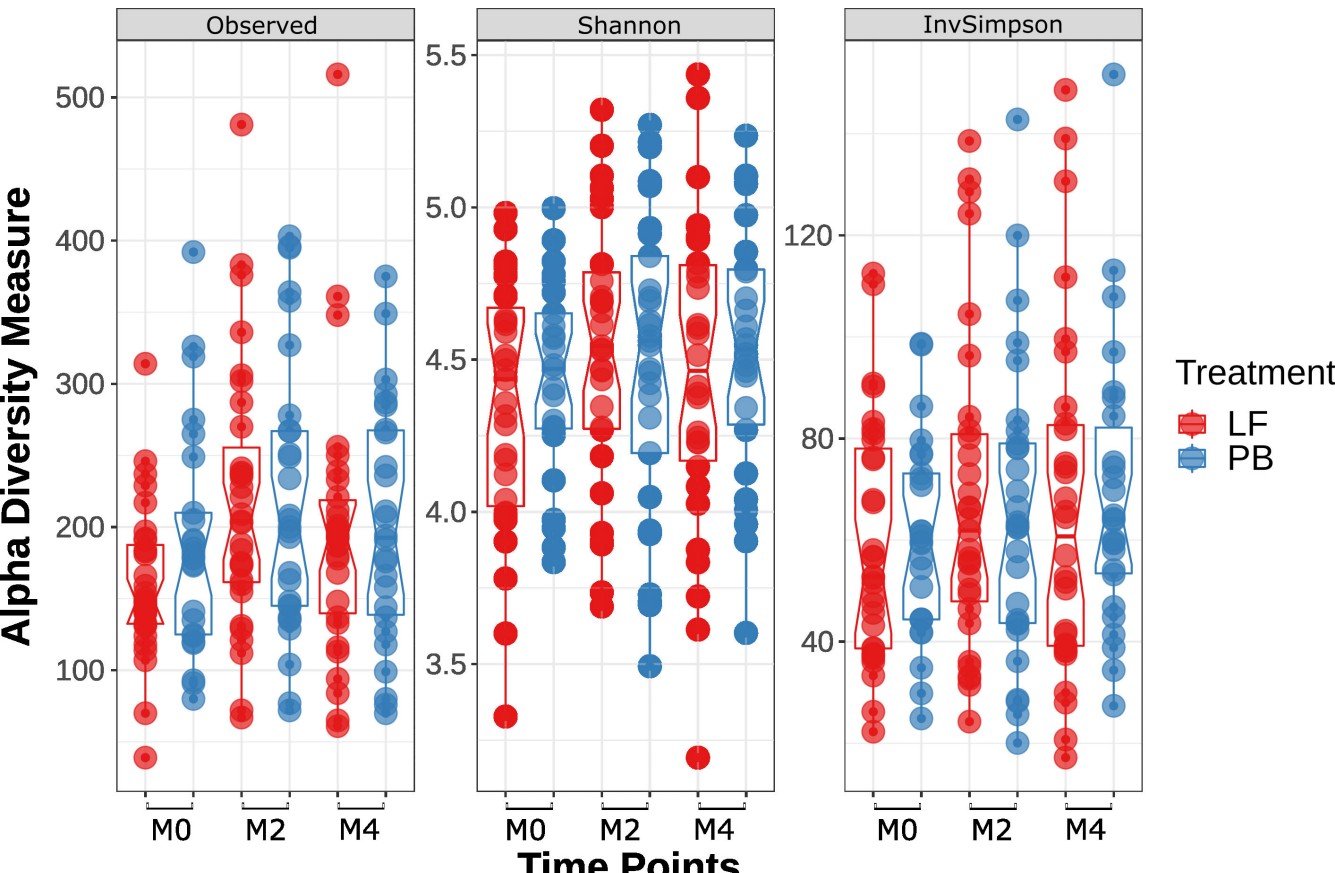

**FIG 2** Alpha diversity stratified by treatment (lactoferrin/placebo) and collection point (M0, M2, and M4). Detailed results and statistics are in the Supplementary Materials section (Table S3).

We found an enrichment of *Christensenella* and *Clostridium* genus in group LF-M4 compared to LF-M0, while no taxa were enriched in group LF-M0 compared to LF-M4 (Fig. 4A). We also observed an enrichment of *Sarcina*, *Christensenella,* and *Coprococcus* genera in the PB-M4 group compared to the PB-M0 group. Conversely, *Clostridium*, *Enterococcus*, *Tyzzerella,* and *Escherichia* were enriched in group PB-M0 compared to group PB-M4 (Fig. 4B). *Bifidobacterium* and *Agathobacter* were increased in group PB-M4 compared to group LF-M4. Furthermore, *Terrisporobacter* and *Bacteroides* were enriched in group LF-M4 compared to PB-M4 (Fig. 4C).

## DISCUSSION

Previous *in vitro* studies have characterized lactoferrin's antimicrobial and bacteriostatic properties (21). Moreover, lactoferrin oral administration has been suggested to enhance diversity in the gut microbiota in neonates (7, 20) while reducing the abundance of opportunistic pathogens (17). However, the effect of lactoferrin on infant gut microbiota over time has not been completely studied, making it difficult to understand its role in early life gut microbiome assembly and therapeutic potential during infancy. To our knowledge, this is the first study to report the effect of lactoferrin on infant gut microbiome composition over time, providing an insight into its probiotic potential in the early life microbiome.

We showed that participants treated with lactoferrin did not show a significant increase in gut microbiome diversity over time than the control group (Fig. 2), which contrasts with previous *in vitro* studies (7, 21). Despite working with a cohort with a more mature gut microbiota compared to previous studies that assessed the impact of lactoferrin in neonates and infants (17, 19), it was challenging to distance the effect of

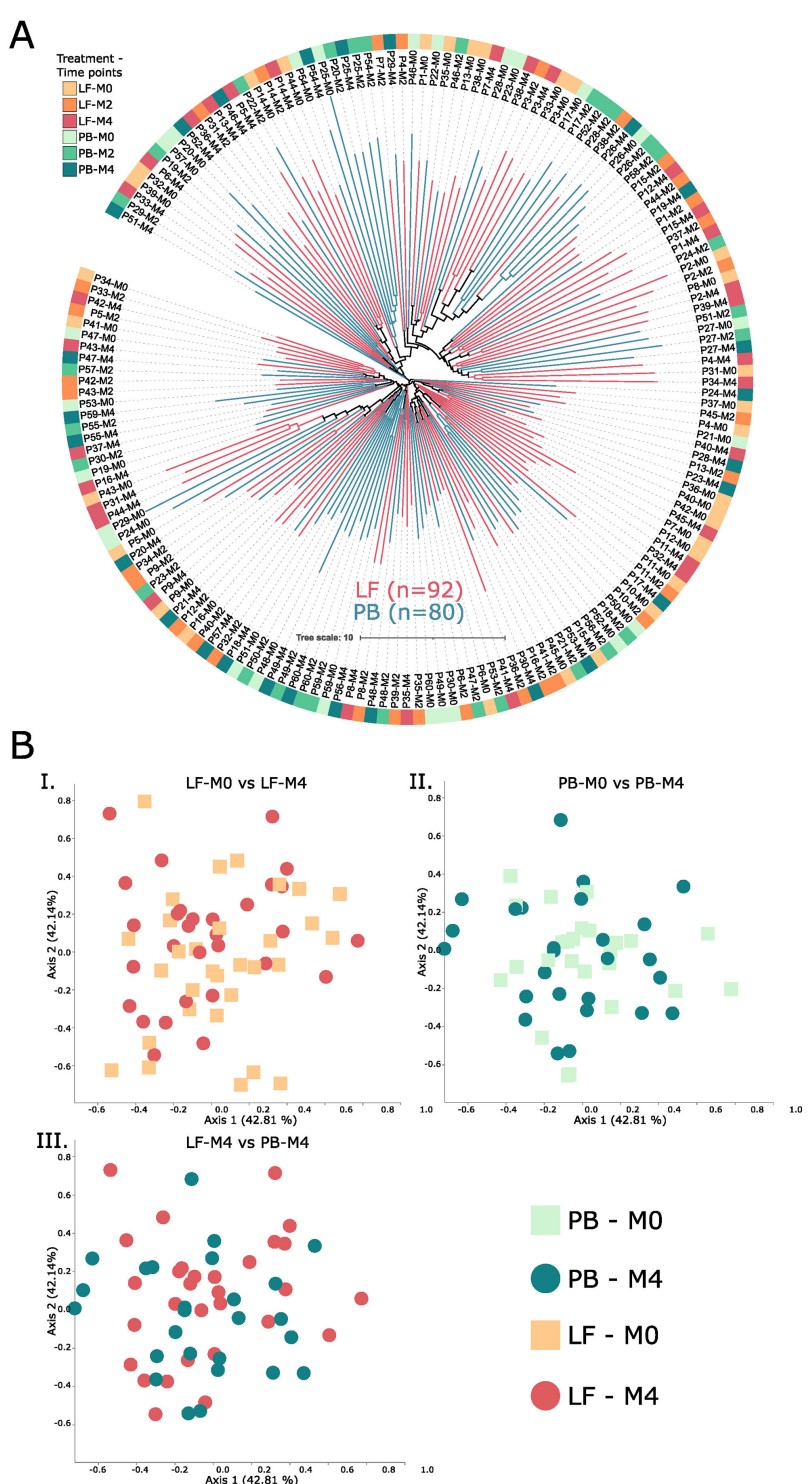

FIG 3 (A) Microbiota composition among experimental groups. Samples were plotted in a dendrogram tree according to their patient correspondence and sampling point (e.g., Patient 8 sample after 2 months of starting the treatment: P8-M2). The outer ring denotes experimental groups (lactoferrin [LF]/placebo [PB]) at three time points. (B) Principal coordinate analysis (PCoA) comparing microbiota composition between experimental groups. We found no difference in overall gut microbiota composition between experimental groups. The supplementary material provides detailed information about the statistical significance (Table S4).

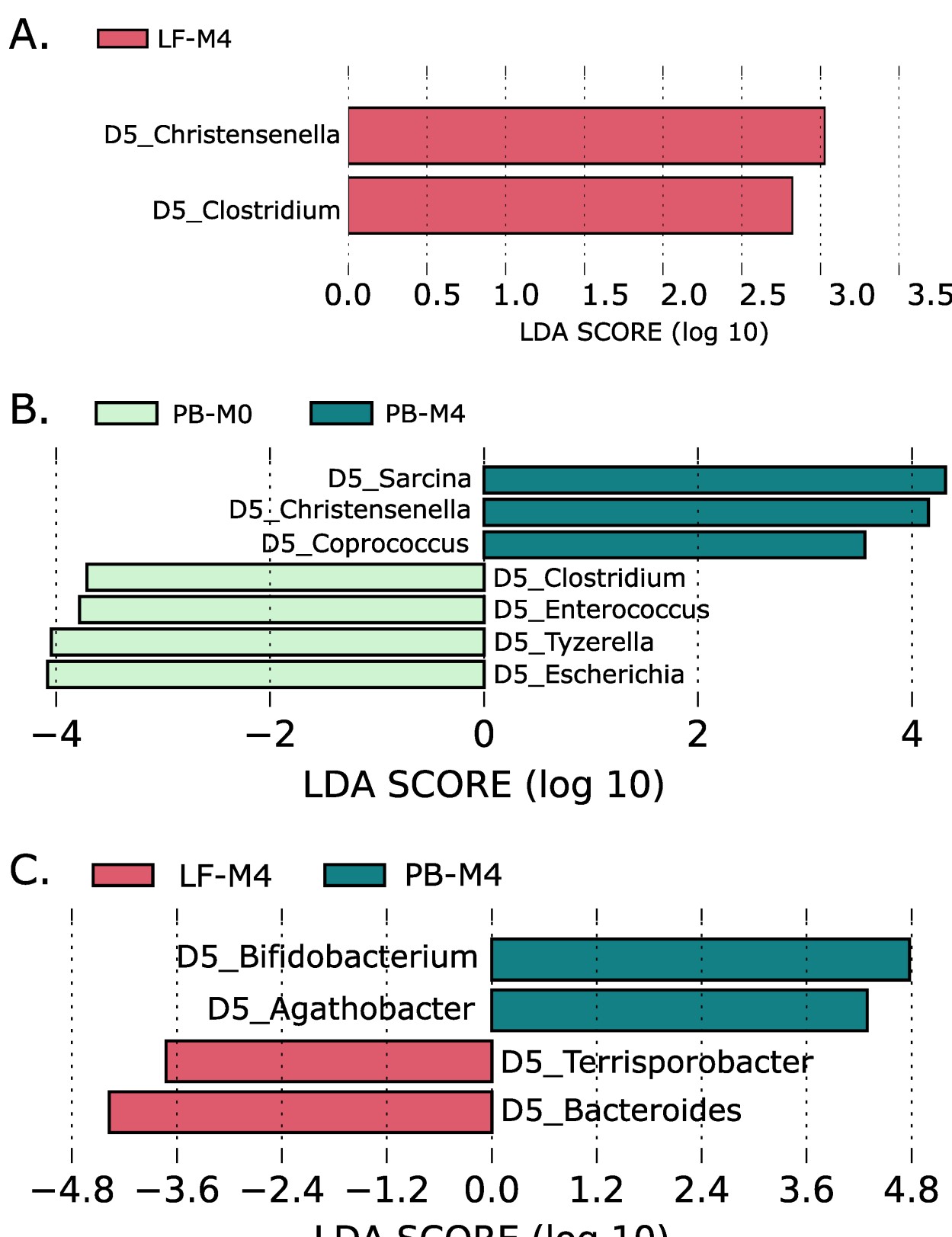

**FIG 4** Significantly enriched bacterial genus across experimental groups: (A) LF-M0 versus LF-M4; (B) PB-M0 versus PB-M4; and (C) LF-M4 versus PB-M4. Detailed information is found in the Supplementary Material section (Table S5).

lactoferrin treatment from normal microbiota maturation in the first years of life since participants aged between 12 and 18 months old when they were enrolled (6).

Samples did not cluster according to the patient's treatment (lactoferrin/placebo), suggesting that lactoferrin does not change the overall gut microbiota composition over time. We also showed that samples were often clustered with those from different patients (Fig. 3). This result was unexpected because samples from the same patient tend to be more similar among them than others (28). Yet, it may be explained by the natural maturation of the infant microbiota since the participants were between 12 and 18 months old at enrollment, making their composition change naturally over time (6). The lack of maturation might explain why samples from the same patient, at different time points, did not cluster together.

We assessed the microbiome composition of groups PB and LF at M0 to ensure that both experimental groups had similar characteristics at the start of the trial. We found an enrichment of *Prevotella* (0.037%), *Megasphaera* (0.001%), and *Roseburia* (0.0011%) in group PB-M0 compared to LF-M0 when performing the LefSE analysis; however, we did not find significant global differences in the microbiome composition between groups when calculating the Aitchison distance. Since the abundance of these species was low compared to the dominant species, and the overall microbiome composition among groups was not significantly different, we assumed that both experimental groups started the trial within the same conditions regarding microbiota composition.

We compared bacterial species enrichment between experimental groups (lactoferrin/placebo) at different time points (M0, M2, and M4) (Fig. 4). When we compared LF-M0 against LF-M4, we found a significant abundance of *Christensenella* in the LF-M4 group. *In vivo* experiments have shown that this genus is correlated with reduced body weight and is one of the most transmitted taxa among relatives (29); however, we cannot attribute the increase of this genus to the lactoferrin treatment since it was also found enriched in the PB-M4 group (compared to the PB-M0 group). This genus has an abundance of 0.18% (+/− 0.58%) in LF-M4 and 0.07% (+/− 0.17%) in PB-M4; however, no statistical difference was found among LF and PB. Hence, the enrichment of *Christensenella* might be caused by gut microbiota maturation over time, independently of treatment. Conversely, we saw an enrichment of the *Clostridium* genus in group LF-M4, usually considered a commensal group in the infant gut microbiota (30); however, this genus also includes *C. difficile,* an important human pathogen (31). Interestingly, *C. difficile* is well tolerated in infants, and the immunoglobulin G antitoxin response produced by the carriage of this bacteria appears to protect subsequence *C. difficile* infections (30). We found that *Coprococcus, Christensenella,* and *Sarcina* genera enriched in the PB-M4 group when comparing it against PB-M0. *Coprococcus* and *Christensenella* genera are probiotic bacteria usually present in infants (32); hence, it makes sense to observe the enrichment of these bacteria along with host maturation. Conversely, *Sarcina* has been reported as a genus of opportunistic pathogens associated with gastric dilatation and gastritis (33). However, there is no information about this genus in the context of the infant gut microbiota. The *Clostridium, Enterococcus, Tyzzerella*, and *Escherichia* genera were enriched in group PB-M0 compared to group PB-M4. Members of these genera are considered commensal bacteria of gut microbiota in infants; however, some species are also clinically relevant pathogens (31, 34–37). We found an enrichment of *Bifidobacterium* and *Agathobacter* in group PB-M4 compared to group LF-M4. *Bifidobacteria* are known probiotics that play an essential role in infant gut microbiome development and correlate with a healthy microbiome status in infants (38, 39). This genus is one of the most abundant species in early life, and it is expected to be enriched in infants (39). Moreover, *Agathobacter* has been described as a probiotic bacterium associated with decreased carbapenemase-producing *Klebsiella pneumoniae* (40) and reduced plasma cholesterol in patients with insulin resistance (41). Moreover, this genus has been described as a marker for good infant health status (42, 43). In contrast, we saw an enrichment of *Bacteroides* and *Terrisporobacter* in group LF-M4 compared to PB-M4. *Bacteroides* are well known for being early colonizers of gut

microbiota and influencing microbiome assembly by creating an anaerobic environment in the gut (44). *Terrisporobacter* has also been enriched in the LF-M4 group; however, there is no information about the role of this group on gut microbiota; hence its clinical impact is unknown.

The main limitation of this study is the immature nature of the gut microbiota in the studied population, which makes it difficult to assess the impact of lactoferrin on gut microbiota since microbiome composition is constantly and naturally shifting during the first years of life. Moreover, the library protocol caused a lot of chimeric sequences (80% on average), causing the loss of most sequences during the quality control process ( Table S2). This problem might be due to PCR artifacts and potential contamination (45); however, the remaining amplicons in samples were sufficient to describe the human microbiota composition ( Fig. S5). Another limitation is the limited statistical power due to the low sample size. Also, samples used in the study had been frozen for almost 10 years, possibly affecting DNA quality. Finally, we did not have information on antibiotic consumption before the start of the treatment. Since antibiotics can disturb the gut microbiota composition for several months (6), we do not know if toddlers presented an undisturbed gut microbiome before enrolling in the original study. The clinical trial from which the samples originated was not designed for microbiome studies. Future early-life microbiome studies should be carefully designed to evaluate the effect of lactoferrin on infant gut microbiota. Individuals could be followed for extended periods until their gut microbiota reach an adult-like state around 3 years of age. They should also consider antibiotic consumption before enrollment and include individuals who do not use antibiotics for at least 6 months before starting the treatment.

In summary, this study suggests that lactoferrin oral treatment in toddlers aged 12–18 months does not affect gut microbiome diversity and composition over time. Despite its limitations, this provides insight into how lactoferrin affects the gut microbiota of young children over time, advancing our knowledge of the effects of this protein on the health of the host and its potential to cure and prevent gut infections.

## MATERIALS AND METHODS

### Study design

Our study is based on a previous community-based, randomized, double-blind controlled trial of bovine lactoferrin to prevent diarrhea in 555 children in Lima-Peru (24). Specimens were collected from July 2008 to May 2009 from infants aged 12–18 months old (Table S1). Children received 0.5 g twice daily lactoferrin or placebo (composed of Maltodextrin, a carbohydrate made from corn starch) diluted in 25 mL of water for 6 months. The dose of lactoferrin was determined based on the estimated amount consumed by a breastfeeding 12-month-old infant (24). Lactoferrin and placebo solutions were mixed with a sugar, a strawberry flavor, and a pink food coloring agent to make the preparation appear and taste identical. Total diarrhea episodes were compared in participants treated with lactoferrin or placebo. Although there was no difference in the number of diarrhea episodes between experimental groups, the infants who received lactoferrin had shorter episodes of diarrhea than the control group (24). This result indicated that lactoferrin treatment positively affected diarrhea prevalence in infants. We excluded samples of children who presented malnutrition and obesity based on the anthropometric data of the study (Table S1). Samples were stored in a −80°C freezer after collection. We selected specimens from 60 infants (31 received lactoferrin and 29 received placebo) at 0, 2 (+/− 2 weeks), and 4 (+/− 2 weeks) months after the beginning of the treatment, accounting for 180 stool samples analyzed.

### Fecal DNA processing and sequencing

Genomic DNA was extracted using a Dneasy PowerSoil Pro Kit (Qiagen, Hilden, Germany) according to manufacturers' instructions and quantified using a Qubit 2.0 Fluorometer

(Life Technologies, Massachusetts, USA). Total genomic DNA was used as a template to amplify the V3–V4 region of the 16S rRNA gene following the standard Illumina protocol (46). We included negative PCR controls in the library preparation to verify nucleic acid contamination. Reads from negative PCR controls did not align with 16S reads from our samples (data not shown). It is known that negative extraction might not be genuinely "negative" due to the bacterial DNA present in DNA extraction kits (28); however, by assuring those reads do not align with the sequencing data from the samples, we can differentiate true sequencing data from low-level amplification from the reagents. Amplicon libraries were normalized to an equimolar concentration and sequenced using an Illumina Miseq instrument with a $2 \times 250$ v2 sequencing kit (Illumina, San Diego, USA).

## Bioinformatic analysis

Barcoded Illumina reads were demultiplexed with bcl2fastq 1.8.4 (47). Human and PhiX sequence contaminations were removed with Bowtie2 2.3.5 (48). Low-quality reads filtering and surviving reads clustered into ASV were done with DADA2 1.14 (49). Samples with less than 1,000 amplicon reads were discarded since they would not successfully describe the gut microbiota composition (50). One hundred seventy-two from 180 patient samples passed QC, generating more than 1,033 amplicons after filtering (range: 1,033–45,349, mean: 8,109) (Table S2). Reads were classified using VSEARCH 2.23 (51) against the Silva 16S database (v132-2018) (52). Data were rarefied to 1,033 amplicons per sample before alpha diversity assessment to normalize the number of reads per sample. Rarefaction was not performed for any other analysis since it would affect the precision of results (53). Alpha diversity was assessed with Phyloseq 3.17 (54) using observed ASVs, Shannon, and the Inverse Simpson diversity index to evaluate richness and evenness among experimental groups (55). Box and whisker plots comparing alpha diversity groups were plotted on ggplot2 3.4.2 (56). The statistical differences between groups were assessed using the Wilcoxon rank-sum test on R 3.6.3. Beta diversity was calculated using an Aitchison distance metric (57) and plotted using a PCoA and a dendrogram. Statistical difference between groups was calculated using the permutational multivariate analysis of variance (PERMANOVA) test using the q2-diversity plug-in for QIIME2 (58). LEfSe 1.0 (Linear discriminant analysis Effect Size) was used to determine specific taxa abundance that differs between experimental groups using the Wilcoxon rank-sum test (27). A table with the percentage of the abundance of genera was provided and transformed into log10 values. Taxa with statistical significance in their abundance were plotted in the LEfSe graph.

## AUTHOR AFFILIATIONS

[1]Laboratorio de Genómica Microbiana, Facultad de Ciencias e Ingeniería, Universidad Peruana Cayetano Heredia, Lima, Peru
[2]Instituto de Medicina Tropical Alexander von Humboldt, Lima, Peru
[3]Facultad de Ciencias Biológicas, Universidad Nacional Mayor de San Marcos, Lima, Peru
[4]Parasites and Microbes Programme, Wellcome Sanger Institute, Hinxton, United Kingdom

## AUTHOR ORCIDs

Luis González http://orcid.org/0000-0002-5776-7179
Pedro E. Romero http://orcid.org/0000-0001-9947-3868
Pablo Tsukayama http://orcid.org/0000-0002-1669-2553

## FUNDING

| Funder | Grant(s) | Author(s) |
|---|---|---|
| Universidad Peruana Cayetano Heredia (UPCH) | | Jose Luis Paredes Sosa |

| Funder | Grant(s) | Author(s) |
|---|---|---|
| ProInnóvate Peru | 289-INNOVATEPERU-EC-2017 | Pablo Tsukayama |

## AUTHOR CONTRIBUTIONS

Luis González, Conceptualization, Data curation, Formal analysis, Investigation, Methodology, Writing – original draft | Jose Luis Paredes Sosa, Conceptualization, Funding acquisition | Susan Mosquito, Conceptualization, Supervision, Writing – review and editing | Yesenia Filio, Investigation, Methodology | Theresa J. Ochoa, Conceptualization, Supervision, Writing – review and editing | Pablo Tsukayama, Conceptualization, Investigation, Supervision, Writing – review and editing.

## DATA AVAILABILITY

The 16s rRNA raw sequences generated during the current study have been deposited under NCBI BioProject PRJNA802048.

## ETHICS APPROVAL

Samples analyzed in this study corresponded to a previous clinical trial. Parents/guardians of participants gave consent authorizing the use of their children's stool samples for future studies. The current study (SIDISI 100261) was approved by the Institutional Ethics Committee of Cayetano Heredia University.

## ADDITIONAL FILES

The following material is available online.

### Supplemental Material

**Supplemental figures (Spectrum00096-23-s0001.svg).** Fig. S1 to S5.
**Table S1 (Spectrum00096-23-s0002.xlsx).** Pediatric information from participants of the study.
**Table S2 (Spectrum00096-23-s0003.xlsx).** Clinical co-variables among treatments (LF/PB).
**Table S3 (Spectrum00096-23-s0004.xlsx).** Clinical co-variables did not affect gut microbiota composition.
**Table S4 (Spectrum00096-23-s0005.xlsx).** Number of surviving sequences after the quality filter.
**Table S5 (Spectrum00096-23-s0006.xlsx).** Alpha diversity metrics stratified by treatment (LF/PB) and collection point (M0, M2, and M4).
**Table S6 (Spectrum00096-23-s0007.xlsx).** Aitchison distance metric among experimental groups.
**Table S7 (Spectrum00096-23-s0008.xlsx).** Comparison of genus abundance stratified by collection point/treatment.

### Open Peer Review

**PEER REVIEW HISTORY (review-history.pdf).** An accounting of the reviewer comments and feedback.

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
