## [Reviewer comments · Microbiology Spectrum]

Microbiology Spectrum

Oral lactoferrin administration does not impact the diversity or composition of the infant gut microbiota in a Peruvian cohort

Luis González, Jose Paredes, Susan Mosquito, Yesenia Filio, Pedro Romero, Theresa Ochoa, and Pablo Tsukayama

Corresponding Author(s): Pablo Tsukayama, Universidad Peruana Cayetano Heredia

Review Timeline:

Submission Date:	January 10, 2023
Editorial Decision:	June 6, 2023
Revision Received:	August 17, 2023
Accepted:	September 5, 2023

Editor: Kevin Theis

Reviewer(s): Disclosure of reviewer identity is with reference to reviewer comments included in decision letter(s). The following individuals involved in review of your submission have agreed to reveal their identity: Zachary D Shaffer (Reviewer #1)

Transaction Report:

DOI: <https://doi.org/10.1128/spectrum.00096-23>

June 6, 2023

Dr. Pablo Tsukayama
Universidad Peruana Cayetano Heredia
Department of Microbiology
Av Honorio Delgado 430
Lima 15102
Peru

Re: Spectrum00096-23 (Oral lactoferrin administration does not impact the diversity or composition of the infant gut microbiota in a Peruvian cohort)

Dear Dr. Pablo Tsukayama:

Thank you for submitting your manuscript to Microbiology Spectrum. The manuscript has been reviewed by two experts in the discipline. As you will see below, some modifications of the manuscript are required before publication. Please let me know if you have any questions or if the attachment with the comments from Reviewer 1 does not come through with this email.

Link Not Available

Sincerely,

Kevin Theis

Journals Department
Reviewer comments:

Reviewer #1 (Comments for the Author):

Major and minor revision suggestions are included in the attached document.

Reviewer #2 (Comments for the Author):

This is a translational study aimed at evaluating whether the administration of lactoferrin to infants affects the alpha and/or beta diversity of their gut microbiota at 12-18 months. The data were obtained from a prior randomized, double-blind controlled trial. Sixty infants received either lactoferrin or placebo twice daily for 6 months. Gut microbiota was characterized from fecal samples collected at 0, 2, and 4 months post-treatment.

The conclusion is that administration of lactoferrin to infants at 0, 2, and 4 months does not affect their microbiota at 12-18 months. The conclusion is justified by the results. Ultimately, this is a study with negative results yet this information is of value. I have only minor comments.

Minor comments:

ABSTRACT

Line 20 - Please consider using lactoferrin throughout, rather than the acronym LF. There is not an issue with manuscript length and the word itself is short. Same for using "PB" for placebo within the text.

Line 21: Use "lactic acid producing" instead of "acid-lactic"

Line 26 - Use "16S rRNA gene" instead of "16S rRNA" (line 47, 129, etc.)

Line 26 - Use "prebiotic" instead of "probiotic"

INTRODUCTION

Line 54 - Change "others" to "other factors"

Line 66 - Use "provides" instead of "plays"

RESULTS

Line 130 - Do the authors have an explanation for why there are such high percentage of chimeras in the sequencing data (see Supplementary Table 2)?

DISCUSSION

Line 241 - Change "bacteria" to "bacterium"

METHODS

Line 296 - How specifically were these negative PCR controls evaluated? Were they sequenced?

Line 327 - Is there a project approval number associated with this institutional ethics committee?

Staff Comments:

Preparing Revision Guidelines

Please return the manuscript within 60 days; if you cannot complete the modification within this time period, please contact me. If you do not wish to modify the manuscript and prefer to submit it to another journal, please notify me of your decision immediately so that the manuscript may be formally withdrawn from consideration by Microbiology Spectrum.

Corresponding authors may join or renew ASM membership to obtain discounts on publication fees. Need to upgrade your

membership level? Please contact Customer Service at Service@asmusa.org.

The manuscript aims to evaluate the long-term changes that lactoferrin administration has on infant microbiome development and whether there are changes in the diversity of the microbiome or enrichment of potentially beneficial microbes.

The **general approach** of the study was to utilize stool samples from a previous study that had treated infants with lactoferrin or placebo over a four-month period and characterize the gut microbiome through DNA sequencing of the 16S rRNA gene amplified from samples collected at 0 months, 2 months, and 4 months, and observe how the gut microbiome structure and composition changes over this period for the different treatment arms.

The **key findings** of the study were that:

- 1) Lactoferrin treatment of infants does not affect the gut microbiome diversity or composition.
- 2) Gut microbiome composition is largely attributable to identity and infant age.
- 3) Lactoferrin treatment is associated with differential enrichment of specific gut microbiome taxa including *Bifidobacterium* and *Bacteroides*.

The **conclusion** of the study was that lactoferrin treatment of infants does not affect gut microbiome diversity or composition longitudinally across a four-month treatment period.

General Comments:

Overall, the introduction and results have good content, but the writing can be improved upon. There are ways that the grammar, layout of the information presented, and sentence structure can be improved that increases the readability of the manuscript. The introduction can also be more specific with regards to the changes that would be expected of early microbiota disruption. The discussion is well written, but the content was lacking and the citations were sparse and did not all seem to be about infant health (especially when discussing the specific taxa). For more content, the authors could pull from the in vitro studies on lactoferrin and some of the taxa that were discussed in this paper if they have been performed. The methods require elaboration on the subject selection, inclusion/exclusion criteria, and how negative controls were dealt with. Other than that, the methods section read fine, and the statistical tests used in the different components of the study are standard for those disciplines. Lastly, consider adding in an analysis on changes in the enterotypes between the treatment and placebo as well.

Major comments:

- 1) The details about the study population and subject criteria are sparse.
- 2) Statistics need to be ran on the subject characteristics at baseline for the LF and PB groups.
- 3) Statistics need to be ran comparing the gut microbiome at M0 for LF and PB.
- 4) Analysis at an enterotype level should be performed.

Minor comments:

Abstract

Comment 1: The abstract has a few typos throughout:

-Line 21: “acid-lactic bacteria” -> “lactic acid bacteria”.

-Line 22: “acid”->”acidic”

-Line 26 “Probiotic”: This term implies that live microorganisms are being given which is not the case in this study.

Comment 2: Line 26-28: Not clear what is being done. Is this an ongoing treatment regimen or single dose of LF?

Comment 3: The abstract is typically a summation of what was done in the paper and therefore future directions is typically not included in the abstract section.

Comment 4: The CDC considers infants to be 0-1 year old. Ensure that the correct term for the age group established by the WHO or other national or international organization is used.

Introduction

Comment 5: Line 62: Rephrase this sentence to make it clear it is the infant that is acquiring the maternal microbiota.

Comment 6: Line 63-66, 72-76. The microbiome is discussed specifically of the vaginally delivered babies that are breastfed. What about those that had C-section, those that were formula fed, and those that were C-section delivered and formula fed? Yes, they are more prone to opportunistic pathogens but what alterations in the microbiota are observed.

Comment 7: Lines 80-82. If transplants work why do this study? Include limitations or risk of fecal transplants as justification for why this study is worth doing.

Comment 8: Line 98: Lactoferrin causes enhanced diversity, but it is never covered whether increased diversity is correlated to good health outcomes.

Methods

Comment 9: Details are lacking on the subject selection and criteria for inclusion and exclusion.

Comment 10: Dosage of LF given was based on breastfeeding infant at 12 months of age but figure 1 makes it seem that no breastfeeding infants were selected. Explain this further.

Comment 11: Did not report where negative control contaminants were removed from the data.

Comment 12: Should include package name and version of all statistical, modeling, and graphical programs and packages used.

Results

Comment 13: Lines 111-119: No mention of age matching treatment and placebo subjects. They should be age matched to reduce confounding variables.

Comment 14: There is no mention of breastfeeding status of the subjects or other inclusion and exclusion criteria here or in the methods. This needs to be included in one of these sections.

Comment 15: Why were only 60 selected when 277 and 278 underwent treatment?

Comment 16: Supplementary table 1: The average BMI, ZWAZ, and ZBMI seem distinct among the groups. The age also mildly differs. Include statistics to show whether there are significant differences in these metrics for the two groups.

Comment 17: Figure 4: How can Clostridium be more enriched in PB-M0 as compared to PB-M4, more enriched in LF-M4 as compared to LF-M0, but not be more enriched in LF-M4 when compared to PB-M4? This doesn't make sense since LF-M0 and PB-M0 should be the same as both are before treatment. Include LF-M0 and PB-M0 comparisons to resolve this issue.

Discussion

Comment 18: Line 200: LF is again referred to as a probiotic. Probiotics typically refers to live microbes being given. LF is not alive.

Comment 19: Line 205-206: The other studies seemed to use neonates and infants as well. It makes more sense to say that the cohort in this study was more mature and in the transition phase away from an immature microbiome.

Comment 20: Line 214-216: Wouldn't the maturation explanation show up in clustering among the respective time points which did not happen either?

Comment 21: Line 219-222: The LF increase and PB increase in *Christensenella* should be presented together since they both increased and were not seen to be different from one treatment to another. Currently is presented a little disingenuously.

Comment 22: Line 226: C. diff is now *Clostridioides difficile*. Also, species name should be lowercase here.

Comment 23: Lines 218-248: Make sure all or at least most of the studies here that are being used to specific taxa's role in the gut microbiome are about infant health.

Spectrum00096-23: Oral lactoferrin administration does not impact the diversity or composition of the infant gut microbiota in a Peruvian cohort) - Response to Reviewers

RESPONSES TO REVIEWER #1

Major comments:

The details about the study population and subject criteria are sparse.

We included further details about the study population and subject criteria for inclusion/exclusion in Lines 146-150.

Statistics need to be run on the subject characteristics at baseline for the LF and PB groups.

We now include a comparison of clinical characteristics at baseline for LF and PB groups (Supplementary Tables 2 and 3). Only BMI was significantly higher in the LF groups (with an average of 16.12 +/- 1.33) compared to the PB groups (with an average of 10.09 +/- 1.32) (Supplementary Table 2); however, its clinical co-variables did not have a significant impact over microbiome composition (Supplementary table 3). Further explanation is provided in lines 158-165 of the revised manuscript.

Statistics need to be run comparing the gut microbiome at M0 for LF and PB.

We compared gut microbiome composition at M0 for LF and PB (Supplementary Figures 3 and 4). We performed a LefSE abundance difference among microbiota populations (Supplementary Figure 3). We found an enrichment of genera *Prevotella*, *Megasphaera*, and *Roseburia* in group PB-M0 compared to LF-M0 (Supplementary Figure 3). However, when calculating the Aitchison distance metric (at M0), we didn't find significant global differences in the microbiome composition between groups (Supplementary Figure 4). Further explanation is described in lines 231-236 of the manuscript.

Analysis at an enterotype level should be performed.

We thank the reviewer for the suggestion. We did not find any dominance nor significant differences in the abundance of the genera *Bacteroides*, *Prevotella*, and *Ruminococcus* among LF and PB, which are the main genera used to describe different types of enterotypes. Moreover, we did not observe any difference in the microbiome composition among experimental groups when calculating the Aitchison distance metric. For these reasons, we decided not to include a formal enterotype analysis.

Minor comments:

ABSTRACT

Comment 1: The abstract has a few typos throughout:

The abstract has been revised and typos have been corrected.

Line 21: “acid-lactic bacteria” -> “lactic acid bacteria”.

The term was changed as suggested.

Line 22: “acid”-> ”acidic”

The term was changed as suggested.

Line 26 “Probiotic”: This term implies that live microorganisms are being given which is not the case in this study.

The term was changed to "prebiotic".

Comment 2: Line 26-28: Not clear what is being done. Is this an ongoing treatment regimen or single dose of LF?

Toddlers received daily oral administration of LF or PB for six months since enrolment. This information is described in line 24 of the manuscript.

Comment 3: The abstract is typically a summation of what was done in the paper and therefore future directions is typically not included in the abstract section.

The description of future directions was removed from the abstract section.

Comment 4: The CDC considers infants to be 0-1 year old. Ensure that the correct term for the age group established by the WHO or other national or international organization is used.

Our participants had more than one year old; hence, we decided to change the term to "toddlers" throughout the manuscript.

INTRODUCTION

Comment 5: Line 62: Rephrase this sentence to make it clear it is the infant that is acquiring the maternal microbiota.

We change "In neonates, part of the maternal microbiota is acquired through" to "Neonates acquire part of the maternal microbiota through" (L64)

Comment 6: Line 63-66, 72-76. The microbiome is discussed specifically of the vaginally delivered babies that are breastfed. What about those that had C-section, those that were formula fed, and those that were C-section delivered and formula fed? Yes, they are more prone to opportunistic pathogens but what alterations in the microbiota are observed.

We added information regarding changes in microbiome composition in infants who had C-section delivery, formula fed, and those who had C-section and were formula fed. More information can be found in L75-82.

Comment 7: Lines 80-82. If transplants work, why do this study? Include limitations or risks of faecal transplants as justification for why this study is worth doing.

Faecal transplant in infants represents a risk since the mother can transfer opportunistic pathogens to the neonate (which has an immature immune system less capable of handling opportunistic colonization). This information is described in lines 107-109 of the manuscript.

Comment 8: Line 98: Lactoferrin causes enhanced diversity, but it is never covered whether increased diversity is correlated to good health outcomes.

A richer microbial diversity is correlated to good health outcomes since it allows the microbiota to have a richer functionality, which the host can take advantage of. This information is described in lines 124-125.

METHODS

Comment 9: Details are lacking on the subject selection and criteria for inclusion and exclusion.

Eligible children were previously weaned at 12-18 months old to evaluate the oral daily dose of lactoferrin's effect in preventing diarrhea exclusively. Exclusion criteria were a history of severe, persistent, or chronic diarrhea, severe malnutrition, serious infections requiring hospitalization the month prior, severe chronic illness, or a personal or family history of allergy to cow's milk or infant formula, eczema, allergic rhinitis, or asthma. This information is described in lines 146-150.

Comment 10: Dosage of LF given was based on breastfeeding infant at 12 months of age but figure 1 makes it seem that no breastfeeding infants were selected. Explain this further.

The original clinical trial (Ochoa et al. 2013) chose an LF dosage concentration based on the concentration of this protein in breastmilk when infants have 12 months old of age to make the dosage close to real life. The original study decided to enrol exclusively weaned infants since they were interested in studying exclusively the impact of LF oral administration in the prevention of diarrhea. Including breastfeeding infants would be problematic since participants would have two sources of LF (breast milk and formula), making it difficult to assess the impact of this protein on diarrhea prevention.

Comment 11: Did not report where negative control contaminants were removed from the data.

We included negative PCR controls in the library preparation to verify nucleic acid contamination. Reads from negative PCR controls did not align with 16S reads from our samples (data not shown). It is known that negative extraction might not be genuinely "negative" due to the bacterial DNA present in DNA extraction kits; however, by assuring those reads do not align with the sequencing data from the samples, we can differentiate true sequencing data from low-level amplification from the reagents. This information is described in lines 453-458.

Comment 12: Should include package name and version of all statistical, modelling, and graphical programs and packages used.

All programs described in the manuscript include information regarding the name and version of all statistical, modelling, and graphical programs and packages used.

RESULTS

Comment 13: Lines 111-119: No mention of age matching treatment and placebo subjects. They should be age matched to reduce confounding variables.

In this study, we selected 60 participants, of which 31 received lactoferrin (15.59 +/- 1.9 months old), and 29 received a placebo (15.93 +/- 2.2 months old). This information is mentioned in lines 155-156.

Comment 14: There is no mention of breastfeeding status of the subjects or other inclusion and exclusion criteria here or in the methods. This needs to be included in one of these sections.

Eligible children were previously weaned at 12-18 months old to evaluate the oral daily dose of lactoferrin's effect in preventing diarrhea exclusively. Exclusion criteria were a history of severe, persistent, or chronic diarrhea, severe malnutrition, serious infections requiring hospitalization the month prior, severe chronic illness, or a personal or family history of allergy to cow's milk or infant formula, eczema, allergic rhinitis, or asthma. This information is described in lines 158-165.

Comment 15: Why were only 60 selected when 277 and 278 underwent treatment?

Most samples were used to cultivate isolates, and most were run out. Also, funds to conduct this study were limited, so we could only perform 16S analysis in 180 samples (corresponding to 60 patients, at three different time points). The selection of patients who underwent PB or LF treatment was randomized.

Comment 16: Supplementary table 1: The average BMI, ZWAZ, and ZBMI seem distinct among the groups. The age also mildly differs. Include statistics to show whether there are significant differences in these metrics for the two groups.

To ensure both experimental groups (LF and PB) started the treatment with the same clinical conditions, we compared the following variables: Weight, BMI, ZWAS, and ZBMI (**Supplementary Table 1**). They were no statistical significance in the clinical covariables among experimental groups, except when comparing the BMI (**Supplementary Table 2**). To assess if these covariables might

interfere with the assessment of lactoferrin over the gut microbiota, we performed a permanova analysis to measure the effects of the clinical covariables in microbiota composition; however, we did not observe a statistically significant impact of these clinical covariables in the gut microbiota (**Supplementary Table 3**). This information can be found in lines 158-165.

Comment 17: Figure 4: How can *Clostridium* be more enriched in PB-M0 as compared to PBM4, more enriched in LF-M4 as compared to LF-M0, but not be more enriched in LF-M4 when compared to PB-M4? This doesn't make sense since LF-M0 and PB-M0 should be the same as both are before treatment. Include LF-M0 and PB-M0 comparisons to resolve this issue.

The group LF-M4 have a higher abundance of *Clostridium* (2.36 +/- 4.54%) compared to group PB-M4 (1.64 +/- 2.97%). However, the difference between both groups was not statistically significant. We performed LefSE analysis to compare the microbiome composition of groups LF and PB at M0 to ensure both experimental groups started with the same conditions at the start of the trial. We found an enrichment of genera *Prevotella*, *Megasphaera*, and *Roseburia* in group PB-M0 compared to LF-M0 (**Supplementary Figure 3**). However, when calculating the Aitchison distance metric, we didn't find significant global differences in the microbiome composition between groups (**Supplementary Figure 4**). Further information can be found in lines 231-236.

DISCUSSION

Comment 18: Line 200: LF is again referred to as a probiotic. Probiotics typically refers to live microbes being given. LF is not alive.

The term was changed to prebiotic as suggested.

Comment 19: Line 205-206: The other studies seemed to use neonates and infants as well. It makes more sense to say that the cohort in this study was more mature and in the transition phase away from an immature microbiome.

Despite working with a cohort with a more mature gut microbiota compared to previous studies that looked at the impact of lactoferrin in neonates and infants, it was challenging to distance the effect of lactoferrin treatment from normal microbiota maturation in the first years of life since participants aged between 12-18 months old when they were enrolled. Further information can be found in lines 273-276.

Comment 20: Line 214-216: Wouldn't the maturation explanation show up in clustering among the respective time points which did not happen either?

The maturation of the gut microbiota in infants is commonly reached at three years old. Participants were 18-24 months old by the end of the clinical trial; hence their microbiota was still immature. The lack of maturation might explain why samples from the same patient (At different time points) did not cluster together. Further information can be found in lines 273-276.

Comment 21: Line 219-222: The LF increase and PB increase in *Christensenella* should be presented together since they both increased and were not seen to be different from one treatment to another. Currently is presented a little disingenuously.

Christensenella has an abundance of 0.18% (+/- 0.58%) in LF-M4 and 0.07% (+/- 0.17%) in PB-M4; however, no statistical difference was found among LF and PB. Further information about this can be found in lines 315-317.

Comment 22: Line 226: *C. diff* is now *Clostridioides difficile*. Also, species name should be lowercase here.

The term was changed as suggested.

Comment 23: Lines 218-248: Make sure all or at least most of the studies here that are being used to specific taxa's role in the gut microbiome are about infant health.

References were changed to refer to taxa's role in the gut microbiome in infant health.

RESPONSES TO REVIEWER #2

Minor comments:

ABSTRACT

Line 20 - Please consider using lactoferrin throughout, rather than the acronym LF. There is not an issue with manuscript length and the word itself is short. Same for using "PB" for placebo within the text.

All acronyms "LF" and "PB" had been changed to "lactoferrin" and "placebo," respectively.

Line 21: Use "lactic acid producing" instead of "acid-lactic"

The term was changed as suggested.

Line 26 - Use "16S rRNA gene" instead of "16S rRNA" (line 47, 129, etc.)

The term was changed as suggested.

Line 26 - Use "prebiotic" instead of "probiotic"

The term was changed as suggested.

INTRODUCTION

Line 54 - Change "others" to "other factors"

The term was changed as suggested.

Line 66 - Use "provides" instead of "plays"

The term was changed as suggested.

RESULTS

Line 130 - Do the authors have an explanation for why there are such high percentage of chimeras in the sequencing data (see Supplementary Table 2)?

This problem might be due to PCR artifacts and possible contamination; however, remaining amplicons in samples were sufficient to describe the human microbiota composition (**Supplementary Figure 5**). (Lines 348 - 350)

DISCUSSION

Line 241 - Change "bacteria" to "bacterium"

The term was changed as suggested.

METHODS

Line 296 - How specifically were these negative PCR controls evaluated? Were they sequenced?

We included negative PCR controls in the library preparation to verify nucleic acid contamination. Reads from negative PCR controls did not align with 16S reads from our samples (data not shown). It is known that negative extraction might not be genuinely "negative" due to the bacterial DNA present

in DNA extraction kits; however, by assuring those reads do not align with the sequencing data from the samples, we can differentiate true sequencing data from low-level amplification from the reagents. (Lines 453-458)

Line 327 - Is there a project approval number associated with this institutional ethics committee?

We have included the project approval code (SIDISI 100261) in the methods section.

September 5, 2023

Dr. Pablo Tsukayama
Universidad Peruana Cayetano Heredia
Department of Microbiology
Av Honorio Delgado 430
Lima 15102
Peru

Re: Spectrum00096-23R1 (Oral lactoferrin administration does not impact the diversity or composition of the infant gut microbiota in a Peruvian cohort)

Dear Dr. Pablo Tsukayama:

Your manuscript has been accepted, and I am forwarding it to the ASM Journals Department for publication. You will be notified when your proofs are ready to be viewed.

Sincerely,

Kevin Theis
Editor, Microbiology Spectrum
